# Full counting statistics as probe of measurement-induced transitions in the quantum Ising chain

Emanuele Tirrito[1,2⋆], Alessandro Santini[1†], Rosario Fazio[2,3] and Mario Collura[1,4‡]

**1** SISSA, Via Bonomea 265, 34136 Trieste, Italy
**2** The Abdus Salam International Center for Theoretical Physics (ICTP),
Strada Costiera 11, I-34151 Trieste, Italy
**3** Dipartimento di Fisica "E. Pancini", Università di Napoli Federico II,
Monte S. Angelo, I-80126 - Napoli, Italy
**4** INFN, Via Bonomea 265, 34136 Trieste, Italy

⋆ etirrito@sissa.it , † asantini@sissa.it , ‡ collura@sissa.it

## Abstract

Non-equilibrium dynamics of many-body quantum systems under the effect of measurement protocols is attracting an increasing amount of attention. It has been recently revealed that measurements may induce different non-equilibrium regimes and an abrupt change in the scaling-law of the bipartite entanglement entropy. However, our understanding of how these regimes appear, how they affect the statistics of local quantities and, finally whether they survive in the thermodynamic limit, is much less established. Here we investigate measurement-induced phase transitions in the Quantum Ising chain coupled to a monitoring environment. In particular we show that local projective measurements induce a quantitative modification of the out-of-equilibrium probability distribution function of the local magnetization. Starting from a GHZ state, the relaxation of the paramagnetic and the ferromagnetic order is analysed. In particular we describe how the probability distributions associated to them show different behaviour depending on the measurement rate.



# 1 Introduction

Isolated many-body quantum systems at zero temperature, governed by an Hamiltonian $H = H_1 + gH_2$ with non-commuting $[H_1, H_2] \neq 0$, may exhibit different phases depending on the value of a physical parameter $g$ appearing in the Hamiltonian. By varying the parameter $g$, quantum fluctuations may drive the many-body ground state across a quantum phase transition [1,2]. This competition between non-commuting operators lies at the heart of quantum mechanics, e.g., inducing correlations and frustration in quantum many-body systems, and forming the cornerstone of quantum technology.

This effect is shared in common with the transition rooted by the the non-commutativity between the generators of the unitary dynamics and the measured operators, which gives rise to macroscopically distinct stationary states.

In fact, recently, the interplay between Hamiltonian, or more generally, unitary evolution and measurements has gained much attention, since monitored quantum systems have been found to undergo a measurement-induced entanglement phase transition [3–5].

In particular, it has been established that quantum systems subjected to both measurements and unitary dynamics offer another class of dynamical behavior described in terms of quantum trajectories [6], and well explored in the context of quantum circuits [7–25], quantum spin systems [26–37], trapped atoms [38], and trapped ions [39–41]. In this context, the bipartite entanglement entropy of isolated systems grows over time and eventually reaches the order of the system size as predicted by the celebrated Cardy-Calabrese quasi-particle picture [42–45]. In this case, the system thermalizes (in a generalised Gibbs sense) and it is characterized by highly entangled eigenstates, i.e. states following an extensive (with the volume) scaling of their entanglement entropy [46–48]. Conversely, projective quantum measurements suppress entanglement growth, such as in the quantum Zeno effect [49–53] according to which continuous projective measurements can freeze the dynamics of the system completely. This question has been addressed in many-body open systems [54–60] whose dynamics is described by a Lindblad master equation [61–63].

However, it has been shown that the average entanglement entropy can still show a transition between a logarithmic and area law phase at a critical measurement strength or to display a purely logarithmic scaling, depending on the stochastic protocol [55]. A logarithmic growth of the entanglement entropy in an entire phase is particularly intriguing, given that the average state is expected to be effectively thermal, and it is reminiscent of a critical, conformally-invariant, phase whose origin has been so far elusive. Similar results have been obtained for free-fermion random circuits with temporal randomness [64], a setting that has been recently generalized to higher dimension [65], or for Majorana random circuits [66,67]. Whether the logarithmic character of the entanglement entropy in the stationary phase survives in the ther-

modynamics limit is still under a very heated debate, since different models and protocols may lead to slightly different conclusions. For example, in Ref. [68] the authors have shown that the average of the stationary entanglement entropy manifests just the area-law behavior, for any finite measurement rate, in the thermodynamic limit. A remnant of a logarithmic scaling is observed only for finite sub-subsystem sizes though, where a characteristic scaling-law between sizes and measurement-rate is established.

In light of these developments, in this work we study the competition between the unitary dynamics and the random projective measurements in a quantum Ising chain coupled to an environment which continuously measures its transverse magnetization. For this particular model several works have discussed the relationship between measurements and entanglement transition. In particular in Refs [30,69,70], the authors considered the one-dimensional quantum Ising model coupled to an environment which continuously measures its transverse magnetization focusing in the quantum state diffusion protocol [71,72] and in the quantum jump [73]. They found a sharp phase transition from a critical phase with logarithmic scaling of the entanglement to an area-law phase. Instead the Ref. [28] presents the transverse Ising model with two non-commuting projective measurements and no unitary dynamics showing the entanglement transition between two distinct steady states that both exhibit area law entanglement.

However, here we want to change back the point of view by restoring the usual connection to the well established way of characterising quantum phase transitions: namely, by identifying a possible local order parameter and by inspecting its full counting statistics.

In particular, we investigate how the stationary probability distribution of the averaged values over the set of quantum trajectories of the magnetizations and its momenta (and cumulant) are affected by the monitoring of local degrees of freedom. In particular, upon increasing the ratio $\gamma$ between measurement rate and Hamiltonian coupling we find a transition from a correlated to a uncorrelated phase, the former characterized by a Gaussian probability distribution of magnetization along the $z$ direction and the latter by a binomial distribution. Moreover, right at the transition point $\gamma_c \simeq 4$, we have numerical evidence that the variance of the probability distribution of the magnetization along $z$ also shows a sharp transition bnetween two different behavior. Indeed, for $\gamma < \gamma_c$, the second $z$-magnetization cumulant grows with the measurement rate; otherwise, for $\gamma > \gamma_c$, it gets bounded. This phase change is get confirmed by the fluctuations of the ferromagnetic correlation function. Where, now, even for relatively small subsystem sizes, we are able to distinguish between two different regimes: from an extensive scaling of the fluctuations with the subsystem sizes, to a vanishing-fluctuating regime.

The content of the manuscript is organised in the following way:

- Sec. 2 is devoted to introduce the model and its description in terms of diagonal spinless fermions and we introduce the Majorana fermions as well.

- In Sec. 3 we discuss the measurement protocols used in our study.

- In Sec. 4 we introduce the formalism beyond mean states and quantum trajectories, stressing the differences between both.

- Sec. 5 collects the main results of our investigation, namely the non-equibrium dynamics generated by a combination of unitary evolving a fully polarised initial state and measurements. After exploring the timedependent behaviour, we mainly focus on the stationary properties. In Sec. 5.1, we analyze the static properties of paramagnetic magnetization for different value of measurement rate $\gamma$. In Sec 5.2 we study the behaviour of the ferromagnetic magnetization in the stationary state.

- Finally, in the Appendices, we collect some details on the Lindbladian dynamics of the averaged states, as well as we focus on the dynamics of the full *quantum probability* dis-

tribution function of the subsystem magnetisation and its connection to the generating function of the moments (and cumulants as well) of the order parameter.

## 2 Model

The Ising Hamiltonian (with no transverse field) reads

$$\hat{H} = -J \sum_{j=1}^{L-1} \hat{\sigma}_j^x \hat{\sigma}_{j+1}^x \,, \tag{1}$$

where $\hat{\sigma}_j^\alpha$ are the local Pauli matrices, such that $[\hat{\sigma}_p^\alpha, \hat{\sigma}_q^\beta] = 2i\delta_{pq}\epsilon^{\alpha\beta\gamma}\hat{\sigma}_p^\gamma$. Here we consider open boundary conditions (OBC). The Hamiltonian is invariant under the action of the global spin flip operator $\hat{P} = \prod_{j=1}^{L} \hat{\sigma}_j^z$. In the following we enforce such symmetry and work in the invariant sector with $P = +1$.

Using the Jordan-Wigner transformation

$$\hat{\sigma}_\ell^x = \prod_{j=1}^{\ell-1}(1-2\hat{n}_j)(\hat{c}_\ell^\dagger + \hat{c}_\ell), \quad \hat{\sigma}_\ell^y = i\prod_{j=1}^{\ell-1}(1-2\hat{n}_j)(\hat{c}_\ell^\dagger - \hat{c}_\ell), \qquad \hat{\sigma}_\ell^z = 1 - 2\hat{n}_\ell, \tag{2}$$

where $\{\hat{c}_i, \hat{c}_j^\dagger\} = \delta_{ij}$ and $\hat{n}_i \equiv \hat{c}_i^\dagger \hat{c}_i$, the Hamiltonian takes the form

$$\hat{H} = -J \sum_{j=1}^{L-1} \left(\hat{c}_j^\dagger - \hat{c}_j\right)\left(\hat{c}_{j+1}^\dagger + \hat{c}_{j+1}\right). \tag{3}$$

Within the approach we will be using in the next sections, it is convenient to replace the fermions $\hat{c}_j$ with the Majorana fermions (here we define two sets of operators through the apexes $x$ and $y$)

$$\hat{a}_j^x = \left(\hat{c}_j^\dagger + \hat{c}_j\right), \qquad \hat{a}_j^y = i\left(\hat{c}_j^\dagger - \hat{c}_j\right), \tag{4}$$

which are hermitian and satisfy the algebra $\{\hat{a}_i^\alpha, \hat{a}_j^\beta\} = 2\delta_{ij}\delta_{\alpha\beta}$, and such that one has

$$\hat{\sigma}_j^x = \prod_{m=1}^{j-1}\left(i\hat{a}_m^y \hat{a}_m^x\right)\hat{a}_j^x, \qquad \hat{\sigma}_j^y = \prod_{m=1}^{j-1}\left(i\hat{a}_m^y \hat{a}_m^x\right)\hat{a}_j^y, \qquad \hat{\sigma}_j^z = i\hat{a}_j^y \hat{a}_j^x. \tag{5}$$

In terms of the Majorana fermions, the Hamiltonian reads

$$\hat{H} = J \sum_{j=1}^{L-1}\left[\frac{i}{2}\hat{a}_j^y \hat{a}_{j+1}^x - \frac{i}{2}\hat{a}_{j+1}^x \hat{a}_j^y\right] = \frac{J}{2}\hat{\mathbf{a}}^\dagger \mathbb{T}\hat{\mathbf{a}}, \tag{6}$$

where we defined the vector $\hat{\mathbf{a}}^\dagger = (\hat{a}_1^x, \ldots, \hat{a}_L^x, \hat{a}_1^y, \ldots, \hat{a}_L^y)$, and identified the $2L \times 2L$ couplings matrix

$$\mathbb{T} = \begin{bmatrix} 0 & \mathbb{H} \\ \mathbb{H}^\dagger & 0 \end{bmatrix}, \tag{7}$$

with $\mathbb{H}_{pq} = -i\delta_{p,q+1}$ for $p,q$ in $\{1,\ldots,L\}$. Introducing the unitary matrix $\mathbb{V} = (v_1, \ldots, v_{2L})$, (i.e. $\mathbb{V}^\dagger \mathbb{V} = \mathbb{I}_{2L\times 2L}$), whose column vectors are parametrised as

$$v_q = \frac{1}{\sqrt{2}}\begin{pmatrix} \phi_q \\ -i\psi_q \end{pmatrix}, \tag{8}$$

we get from the eigenvalue equation $\mathbb{T}v_q = \epsilon_q v_q$ the following coupled equations

$$-i\mathbb{H}\psi_q = \epsilon_q\phi_q, \tag{9}$$

$$\mathbb{H}^\dagger\phi_q = -i\epsilon_q\psi_q. \tag{10}$$

We can notice here that these equations are invariant under the simultaneous change $\epsilon_q \to -\epsilon_q$ and $\psi_q \to -\psi_q$. So, to each positive eigenvalue, $\epsilon_q > 0$, corresponds a negative eigenvalue $\epsilon_{q'} = -\epsilon_q$ with the associated eigenvector $v_{q'} = (\sigma^z \otimes \mathbb{I}_{L \times L})v_q$. From these equations it is straightforward to obtain two decoupled eigenvalue equations $\mathbb{H}\mathbb{H}^\dagger\phi_q = \epsilon_q^2\phi_q$ and $\mathbb{H}^\dagger\mathbb{H}\psi_q = \epsilon_q^2\psi_q$.. Since $\mathbb{H}\mathbb{H}^\dagger$ and $\mathbb{H}^\dagger\mathbb{H}$ are real symmetric matrices, their eigenvectors can be chosen real and they satisfy completeness and orthogonality relations.

In the specific case of the Ising Hamiltonian in Eq. (6), those matrices are already diagonal (with one eigenvalue equals to zero, and $L-1$ eigenvalues equal to one), specifically $(\mathbb{H}\mathbb{H}^\dagger)_{pq} = \delta_{pq} - \delta_{p1}\delta_{q1}$ and $(\mathbb{H}^\dagger\mathbb{H})_{pq} = \delta_{pq} - \delta_{pL}\delta_{qL}$. Choosing the coefficients $\phi_{pq} = \delta_{pq}$ for $p$ and $q$ in $\{1,\dots L\}$, leads to $\psi_{pq} = -\delta_{p\,q-1}$ for $q$ in $\{2,\dots,L\}$ and $\psi_{p1} = -\delta_{pL}$. This implies $\mathbb{V}^\dagger\mathbb{T}\mathbb{V} = \sigma^z \otimes \mathbb{H}\mathbb{H}^\dagger$. From the Majorana field we get the following diagonal Fermi operators corresponding to positive energies

$$\hat{\eta}_q = \frac{1}{2}\sum_{p=1}^{L}\left[\phi_{pq}\hat{a}_p^x + i\psi_{pq}\hat{a}_p^y\right] = \frac{1}{2}\left[\hat{a}_q^x - i\hat{a}_{q-1}^y\right], \quad \text{for} \quad q = 2,\dots,L, \tag{11}$$

and $\hat{\eta}_1 = [\hat{a}_1^x - i\hat{a}_L^y]/2$. They satisfy canonical anticommutation relations $\{\hat{\eta}_q, \hat{\eta}_p^\dagger\} = \delta_{pq}$. From those, the inverse relations reads

$$\hat{a}_q^x = \hat{\eta}_q + \hat{\eta}_q^\dagger, \qquad \hat{a}_q^y = i[\hat{\eta}_{q+1} - \hat{\eta}_{q+1}^\dagger], \tag{12}$$

with $\hat{a}_L^y = i[\hat{\eta}_1 - \hat{\eta}_1^\dagger]$, leading to the diagonal Hamiltonian

$$\hat{H} = \sum_{q=1}^{L}\epsilon_q\,\hat{\eta}_q^\dagger\hat{\eta}_q - J(L-1), \tag{13}$$

with $\epsilon_p = 2J(1-\delta_{p1})$. From the previous relations, the unitary time evolution of the Majorana operators can be easily worked out

$$\hat{a}_p^x(t) = \cos(\epsilon_p t)\hat{a}_p^x - \sin(\epsilon_p t)\hat{a}_{p-1}^y, \tag{14}$$

$$\hat{a}_p^y(t) = \sin(\epsilon_{p+1} t)\hat{a}_{p+1}^x + \cos(\epsilon_{p+1} t)\hat{a}_p^y, \tag{15}$$

where periodic boundary conditions in the indices are intended, namely $0 \to L$ and $L+1 \to 1$.

For a Gaussian state all the information is encoded in the two-point correlation function of the Majorana operators, namely

$$\mathbb{A} = \langle\hat{\mathbf{a}} \cdot \hat{\mathbf{a}}^\dagger\rangle = \begin{pmatrix} \mathbb{A}^{xx} & \mathbb{A}^{xy} \\ \mathbb{A}^{yx} & \mathbb{A}^{yy} \end{pmatrix}, \tag{16}$$

which under the classical Ising Hamiltonian evolve from time $s$ to time $s+t$ according to $\mathbb{A}(s+t) = \mathbb{R}(t)\mathbb{A}(s)\mathbb{R}^\dagger(t)$, with

$$\mathbb{R}(t) = \begin{pmatrix} \mathbb{R}^{xx}(t) & \mathbb{R}^{xy}(t) \\ \mathbb{R}^{yx}(t) & \mathbb{R}^{yy}(t) \end{pmatrix}, \tag{17}$$

whose matrix elements are

$$R_{pq}^{xx}(t) = \cos(\epsilon_p t)\delta_{pq}, \tag{18a}$$

$$R_{pq}^{yy}(t) = \cos(\epsilon_{p+1} t)\delta_{pq}, \tag{18b}$$

$$R_{pq}^{yx}(t) = \sin(\epsilon_{p+1} t)\delta_{p\,q-1}, \tag{18c}$$

$$R_{qp}^{xy}(t) = -\sin(\epsilon_p t)\delta_{p\,q+1}. \tag{18d}$$

# 3 Protocol

We prepare the system in the symmetric ($P = +1$) ground state of the Hamiltonian in Eq. (1), namely the GHZ state

$$|\psi_0\rangle = \frac{1}{\sqrt{2}} \left[ |\cdots\uparrow\cdots\rangle + |\cdots\downarrow\cdots\rangle \right], \tag{19}$$

where here $|\uparrow\rangle$ and $|\downarrow\rangle$ represents the eigenstates of $\hat{\sigma}^x$ with eigenvalues respectively $+1$ and $-1$. This initial state is described by a correlation matrix whose matrix elements are

$$\mathbb{A}_{pq}^{xx} = \mathbb{A}_{pq}^{yy} = \delta_{pq}, \qquad\qquad \mathbb{A}_{pq}^{xy} = -i\delta_{pq+1}, \qquad\qquad \mathbb{A}_{pq}^{yx} = +i\delta_{p+1q}, \tag{20}$$

where once again, PBC in the indices are intended, i.e. $L + 1 \rightarrow 1$; as expected, the initial correlation matrix would be unaffected by just the unitary evolution generated by the Ising Hamiltonian in Eq. (1). However, the system experiences random interactions with local measuring apparatus such that the full time-dependent protocol becomes highly non-trivial. In practice, with a characteristic rate $\gamma$, for each single lattice site $k$, the local magnetization along $\hat{z}$ is measured, i. e. $\hat{\sigma}_k^z = \sum_\sigma \sigma \hat{P}_k^{(\sigma)}$. Here $\sigma = \pm 1$ are the possible outcomes of the measurements, and $\hat{P}_k^{(\sigma)} = (1 + \sigma\hat{\sigma}_k^z)/2$ is the projector to the corresponding subspace.

Let us stress that both the unitary evolution and the local projective measurements keep the state Gaussian in terms of the Majorana fermions. While the former comes straightforwardly from the fact that $\exp(-it\hat{H})$ is Gaussian; the latter may not be immediately visible from the simple structure of the projectors $\hat{P}_k^\sigma$. However, it is easy to show that

$$\hat{P}_k^{(\sigma)} = \lim_{x\to\infty} \frac{e^{x\sigma\hat{\sigma}_k^z}}{\mathrm{Tr}(e^{x\sigma\hat{\sigma}_k^z})}, \tag{21}$$

thus also being a Gaussian operator in terms of Majorana fermions. Finally, let us mention that the protocol also preserves the spin-flip invariance, the state thus remaining always in the $P = +1$ sector.

For the aforementioned reasons, during the entire dynamics, the full information of the state is completely encoded within the two-point functions $\mathbb{A}_{pq}^{\alpha\beta} = \langle \hat{a}_p^\alpha \hat{a}_q^\beta \rangle$, and all higher-order correlators slipt into sums of products of the two-point function only, according to the Wick theorem.

Since $\hat{\sigma}_k^z$ operators acting on different lattice sites commute, we can measure the local spins in any arbitrary order; specifically, if at time $t$ the $k$-th site has been measured, following the Born rule, if the outcome is $\sigma = \pm 1$, then the state $|\Psi(t)\rangle$ transforms into $\hat{P}_k^{(\sigma)}|\Psi(t)\rangle / \sqrt{\langle\Psi(t)|\hat{P}_k^{(\sigma)}|\Psi(t)\rangle}$. The resulting state remaining Gaussian, we can thus focus on the two-point function $\mathbb{A}_{pq}^{\alpha\beta}(t)$ which completely characterises the entire system. The recipe is the following: for each time step $dt$ and each site $k$, we extract a random number $q_k \in (0, 1]$ and only if $q_k \leq \gamma dt$ we take the measurement of $\sigma_k^z$. In such case, we extract another random number $p_k \in (0, 1]$, and the two-point function immediately after the projection to the $\hat{\sigma}_k^z$ local eigenstates becomes (in the following we omit the time dependence in order to simplify the notation)

$$\mathbb{A}_{pq}^{\alpha\beta}\Big|_\sigma = \frac{2}{1 + i\sigma\mathbb{A}_{kk}^{yx}} \left[ \frac{1}{4}\mathbb{A}_{pq}^{\alpha\beta} + \frac{i\sigma}{4}\langle\{\hat{a}_p^\alpha\hat{a}_q^\beta, \hat{a}_k^y\hat{a}_k^x\}\rangle - \frac{1}{4}\langle\hat{a}_k^y\hat{a}_k^x\hat{a}_p^\alpha\hat{a}_q^\beta\hat{a}_k^y\hat{a}_k^x\rangle \right], \tag{22}$$

where $\sigma = +1$ if $p_k \leq 1/2 + \langle\hat{\sigma}_k^z\rangle/2$, otherwise $\sigma = -1$.

The second term can be easily evaluated using the Wick theorem obtaining

$$\langle\{\hat{a}_p^\alpha\hat{a}_q^\beta, \hat{a}_k^y\hat{a}_k^x\}\rangle = 2\mathbb{A}_{kk}^{yx}\mathbb{A}_{pq}^{\alpha\beta} + \left(\mathbb{A}_{pk}^{\alpha x}\mathbb{A}_{qk}^{\beta y} + \mathbb{A}_{kp}^{x\alpha}\mathbb{A}_{kq}^{y\beta}\right) - \left(\mathbb{A}_{pk}^{\alpha y}\mathbb{A}_{qk}^{\beta x} + \mathbb{A}_{kp}^{y\alpha}\mathbb{A}_{kq}^{x\beta}\right). \tag{23}$$

Finally, using the fact that

$$\hat{a}_p^\alpha \hat{a}_q^\beta \hat{a}_k^y \hat{a}_k^x = -4\delta_{pk}\delta_{qk}\delta^{\alpha y}\delta^{\beta x} + 2\delta_{qk}\delta^{\beta y}\hat{a}_p^\alpha \hat{a}_k^x + 2\delta_{pk}\delta^{\alpha y}\hat{a}_k^x \hat{a}_q^\beta \tag{24}$$

$$+ 2\delta_{qk}\delta^{\beta x}\hat{a}_k^y \hat{a}_p^\alpha - 2\delta_{pk}\delta^{\alpha x}\hat{a}_k^y \hat{a}_q^\beta + \hat{a}_k^y \hat{a}_k^x \hat{a}_p^\alpha \hat{a}_q^\beta \,, \tag{25}$$

after a bit of algebra, also the last term in Eq. (22) can be explicitly decomposed as follow

$$\langle \hat{a}_k^y \hat{a}_k^x \hat{a}_p^\alpha \hat{a}_q^\beta \hat{a}_k^y \hat{a}_k^x \rangle = -\mathbb{A}_{pq}^{\alpha\beta} - 4\delta_{pk}\delta_{qk}\left(\delta^{\alpha y}\delta^{\beta x} - \delta^{\beta y}\delta^{\alpha x}\right)\mathbb{A}_{kk}^{yx} - 2\delta_{qk}\delta^{\beta y}\mathbb{A}_{kp}^{y\alpha} +$$
$$+ 2\delta_{pk}\delta^{\alpha y}\mathbb{A}_{kq}^{y\beta} - 2\delta_{qk}\delta^{\beta x}\mathbb{A}_{kp}^{x\alpha} + 2\delta_{pk}\delta^{\alpha x}\mathbb{A}_{kq}^{x\beta} \,. \tag{26}$$

## 4  Mean state and quantum trajectories

In this work, we study observables affected by the continuous monitoring of the system. Before doing so, it is important to stress the differences between quantum trajectories and mean states [54]. The mean state of our protocol is defined as the average of the density matrix over the measurements outcomes

$$\overline{\hat{\rho}_t} = \overline{|\psi_t\rangle\langle\psi_t|}\,, \tag{27}$$

where with $\overline{(\dots)}$ we denote the average over the measurement protocol. The Lindblad master equation associated to our protocol, which describes the time-evolution of the mean state, is given by

$$\partial_t\hat{\rho} = -i[\hat{H},\hat{\rho}] + \frac{\gamma}{2}\sum_{k=1}^{L}\left(\hat{\sigma}_k^z \hat{\rho} \hat{\sigma}_k^z - \frac{1}{2}\{\hat{\sigma}_k^z\hat{\sigma}_k^z,\hat{\rho}\}\right)\,, \tag{28}$$

see Appendix A for its derivation. Since the evolution is implemented by an unital dynamical quantum map, then the completely mixed state is a fixed point of the dynamics. We therefore expect the dynamics to bring the mean state (apart from symmetry protected sectors of the Hilbert space) toward the trivial infinite temperature one. Therefore, we say that averages computed with the mean state are known *a priori*.

On the other hand, we may consider single quantum trajectories described by a set of not-averaged density matrices $\hat{\rho}_{t,\xi} = |\psi_{t,\xi}\rangle\langle\psi_{t,\xi}|$ where $\xi$ represents a single realization of the stochastic protocol. We then consider averages of a functional of our state $\mathcal{F}[\hat{\rho}]$ over the set of quantum trajectories, it is apparent that

$$\mathcal{F}[\overline{\hat{\rho}_t}] \neq \overline{\mathcal{F}[\hat{\rho}_{t,\xi}]}\,, \tag{29}$$

as long as $\mathcal{F}$ is *not a linear* functional of $\hat{\rho}_{t,\xi}$. As a simple example we observe that the purity of our states $\overline{\text{Tr}\{\hat{\rho}_{t,\xi}^2\}} = 1$ for the set of quantum trajectories (since the state is always a product state), meanwhile since the mean state is generically mixed we have $\text{Tr}\{\overline{\hat{\rho}_t}^2\} < 1$.

Let us now consider an operator $\hat{A}$ and a set of quantum trajectories $\hat{\rho}_{t,\xi}$. Given a certain fixed realization of the measurement protocol $\xi$ we can define a *quantum probability*

$$\mathscr{P}_{t,\xi}(a;\hat{A}) = \text{Tr}\{\delta(\hat{A}-a)\hat{\rho}_{t,\xi}\} \tag{30}$$

of obtaining certain outcomes from the eigenvalues of $\hat{A}$. Given that this distribution is linear in $\rho_{t,\xi}$, following the previous discussion, we have that the average of the distribution over the set of quantum trajectories

$$\mathscr{P}_t(a;\hat{A}) = \overline{\mathscr{P}_{t,\xi}(a;\hat{A})} = \text{Tr}\{\delta(\hat{A}-a)\overline{\hat{\rho}_{t,\xi}}\} \tag{31}$$

is a deterministic quantity known *a priori*, which is completely described by the dynamics of the mean state. Furthermore, all the moments of $\mathscr{P}_{t,\xi}(a)$, i.e.

$$\left\langle \hat{A}_{t,\xi}^n \right\rangle = \text{Tr}\{\hat{A}^n \hat{\rho}_{t,\xi}\} \tag{32}$$

are linear functionals of $\hat{\rho}_{t,\xi}$ and therefore display a deterministic *a priori* dynamics. Despite this we can consider the cumulants of the distributions (30) over the set of quantum trajectories which result in *non-linear* functional of $\rho_{t,\xi}$. In particular, the $n$-th cumulants of the distribution is given by

$$K_{t,n}(\hat{A}) = \partial_\lambda^n \overline{\log\left[\text{Tr}\{e^{\lambda \hat{A}} \hat{\rho}_{t,\xi}\}\right]}\Big|_{\lambda=0}. \tag{33}$$

As for instance, we may write the second cumulant

$$K_{t,2}(\hat{A}) = \overline{\text{Tr}\{\hat{A}^2 \hat{\rho}_{t,\xi}\}} - \overline{\text{Tr}\{\hat{A}\hat{\rho}_{t,\xi}\}^2} = \text{Tr}\{\hat{A}^2 \overline{\hat{\rho}_{t,\xi}}\} - \overline{\text{Tr}\{\hat{A}\hat{\rho}_{t,\xi}\}^2}, \tag{34}$$

which is clearly given by an average of a *non-linear* functional of $\rho_{t,\xi}$.

We are going now to construct a different probability distribution whose second moment is the very same non-linear contribution of the former cumulant $\overline{\text{Tr}\{\hat{A}\hat{\rho}_{t,\xi}\}^2}$. Indeed, we may consider a *classical probability* obtained by considering $\mathscr{N}$ different trajectories and computing the average of the observable over each realization of the stochastic protocol $\xi$

$$a_{t,\xi} = \text{Tr}\{\hat{A}\hat{\rho}_{t,\xi}\}, \tag{35}$$

in the limit of $\mathscr{N} \to \infty$ the averages over this set will be distributed according to a probability distribution

$$P_t(a;\hat{A}) = \lim_{\mathscr{N} \to \infty} \frac{1}{\mathscr{N}} \sum_{\xi=1}^{\mathscr{N}} \delta(a_{t,\xi} - a) = \overline{\delta\left(\text{Tr}\{\hat{A}\hat{\rho}_{t,\xi}\} - a\right)}, \tag{36}$$

not dependent on the particular realization $\xi$ and *non-linear* in $\rho_{t,\xi}$. Then, we can consider the moments of the latter distribution

$$\mu_{t,n}(\hat{A}) = \int P_t(a;\hat{A}) a^n \, da, \tag{37}$$

which in this case are *non-linear* functionals of $\rho_{t,\xi}$. As a clarifying example, let us consider the second moment

$$\mu_{t,2}(\hat{A}) = \int P_t(a;\hat{A}) a^2 \, da = \lim_{\mathscr{N} \to \infty} \frac{1}{\mathscr{N}} \sum_{\xi=1}^{\mathscr{N}} \left[\text{Tr}\{\hat{A}\hat{\rho}_{t,\xi}\}\right]^2 = \overline{\text{Tr}\{\hat{A}\hat{\rho}_{t,\xi}\}^2}. \tag{38}$$

We notice that this latter probability distribution contains the information on all the moments $\overline{\left\langle \psi_\xi \middle| \hat{A} \middle| \psi_\xi \right\rangle^n}$, which describe the statistical properties of the average of $\hat{A}$ over the set of quantum trajectories.

As it is apparent, there is a close connection among the cumulants of $\mathscr{P}_{t,\xi}(a;\hat{A})$ and the moments of the distributions $\{P_t(a;\hat{A}^k)\}_{k\in\mathbb{N}}$. This is due to the fact that it is possible to compute the $n$-th cumulant of $\mathscr{P}_{t,\xi}(a;\hat{A})$ with a linear combination the moments of $\{P_t(a;\hat{A}^k)\}_{k\leq n}$.

In order to examine the melting of the ferromagnetic order of the Ising chain under continuous projective paramagnetic measures, we will consider the following observables

$$\hat{M}_\ell^z = \frac{1}{2}\sum_{j\in\ell} \hat{\sigma}_j^z, \qquad \hat{M}_\ell^x = \frac{1}{2}\sum_{j\in\ell} \hat{\sigma}_j^x, \qquad \hat{M}_\ell^{xx} = \frac{1}{4}\sum_{i\neq j\in\ell} \hat{\sigma}_i^x \hat{\sigma}_j^x, \tag{39}$$

and study the classical distribution $P_t$ of the averages computed over a set quantum trajectories or, when this will not be possible, the cumulants of $\mathscr{P}_{t,\xi}$.

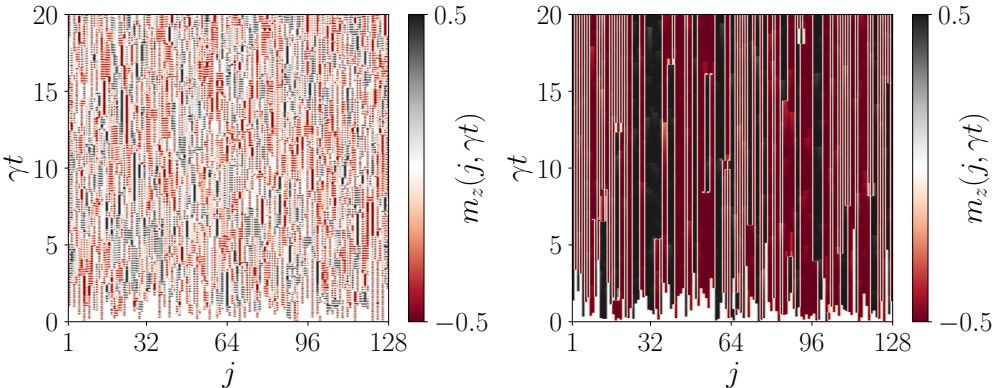

Figure 1: Local $z$-magnetization computed on single realization of a quantum trajectory. Left panel: $\gamma = 0.1$; Right panel: $\gamma = 10$.

## 5 Numerical results

We recap here the numerical procedure that implements the continuous measurement protocol. We remark that, since we are working with an evolution that preserves the state Gaussian, the correlation matrix $\mathbb{A}$ contains all the information of the system. The starting point of the dynamics is the GHZ state whose correlation matrix is given in Eq. (20). The system is then evolved unitarily by $dt$ with Eqs. (18), then we apply the projective measurement step. To do so, sequentially projective measurements of the $z$-magnetization on each site are applied with probability $p_{\text{meas}} = \gamma dt$, thus transforming the system correlation matrix $\mathbb{A}$ as pointed out in Eq. (22).

In our simulations, in order to set a time scale, we evolve our system up to a fixed time which depends on gamma, we chose $t_f = \mathcal{T}/\gamma$. This means that, on average, for each choice of $\gamma$ the same number of projective measurements are executed. Indeed, we have $t_f/dt$ time steps where with probability $\gamma dt$ for each of the $L$ sites a projective measurement is done. This implies an average number of measurements of

$$N_{\text{meas}} = \frac{t_f}{dt} L \gamma dt = \mathcal{T} L \,, \tag{40}$$

in our simulations $\mathcal{T} = 20$, $L = 128$ then, on average, each realization of the stochastic protocol consist of $N_{\text{meas}} = 2560$ projective measurements. Furthermore, in the following, for each choice of the parameters, we chose a set of $\mathcal{N} = 200$ quantum trajectories.

In the following paragraphs, we study how the initial ferromagnetic order melts under the influence of repeated measures.

### 5.1 Paramagnetic magnetization

First of all, we start by analyzing the dynamics of the paramagnetic magnetization, we will denote with $|0\rangle$ and $|1\rangle$ the two eigenstates of $\hat{\sigma}^z$ with eigenvalue 1 and −1 respectively. In Fig. 1 we show the evolution of the local $z$-magnetization

$$m_z(j, \gamma t) = \frac{1}{2} \text{Tr}\left\{\hat{\sigma}_j^z \hat{\rho}_{t,\xi}\right\}, \tag{41}$$

for a single realization of the stochastic protocol and two different choices of the measurement rate $\gamma$. It is apparent that, due to the quantum Zeno effect [50, 51], increasing the measurement rate, local regions in which the magnetization is frozen appear. On the other hand, if measurements are sparse in time we expect a completely random evolution of the system.

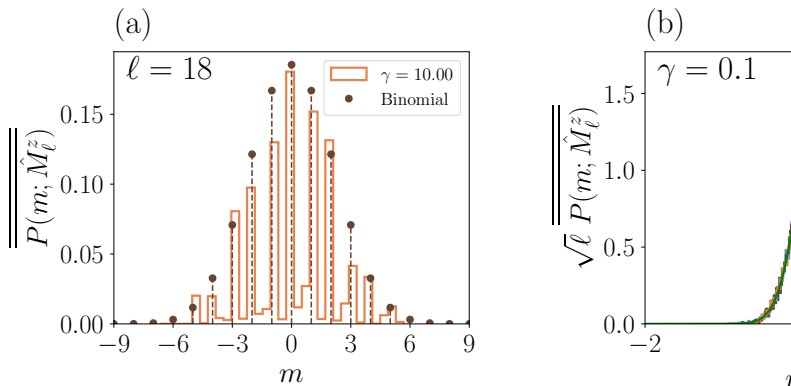

Figure 2: Stationary probabilities of the subsystem paramagnetic magnetization. (a) regime of fast measurements $\gamma = 10$, the numerically probability (obtained from an histogram) is compared to a binomial distribution. (b) sparse measurements regime $\gamma = 0.1$ the numerically probability (obtained from an histogram) is compared to a Gaussian distribution. Details on the normalization of the two histograms are in the main text.

To be more quantitative, we are going to analyze the behavior of $P_t\left(m_\ell^z; \hat{M}_\ell^z\right)$, which we remind is the probability distribution of the averaged value of the subsystem paramagnetic magnetization over the set of quantum trajectories, in the stationary case for which $\gamma t \gg 1$ by defining the distribution

$$\overline{\overline{P(m; \hat{M}_\ell^z)}} = \frac{1}{t_f - t_0} \int_{t_0}^{t_f} P_t\left(m_\ell^z; \hat{M}_\ell^z\right) \mathrm{d}t \,, \tag{42}$$

where $\overline{\overline{(\dots)}}$ denotes the time average, in our simulations we chose $t_0$ such that $\gamma t_0 = 5$, we will study the aforementioned limiting case of fast measurements $\gamma \gg 1$ and rare $\gamma \ll 1$.

We start our analysis by the limit case in which $\gamma \gg 1$ we are constantly monitoring all the sites of the system, the unitary evolution thus becomes negligible, and therefore we are effectively blocking the system in the product state outcome of first measurement. Since we are starting from the ferromagnetic GHZ ground state, the first measurement outcome with equal probability is one of the product states $|\tau_1 \dots \tau_L\rangle$, with $\tau_j = 0, 1$ for $j = 1, ..., L$. Since in this limit the state is blocked in the first measurement outcome we have that $\overline{\overline{P(m_\ell^z)}}$ will be equivalent to the quantum probability $\mathscr{P}(m_\ell^z; \hat{M}_\ell^z)$ of obtaining from the state a certain eigenvalue of $\hat{M}_\ell^z$. Thus $\overline{\overline{P(m_\ell^z)}}$ will be the discrete binomial distribution

$$\overline{\overline{P(m; \hat{M}_\ell^z)}} = \frac{1}{2^\ell} \binom{\ell}{m_\ell^z + \frac{\ell}{2}}, \qquad m_\ell^z \in -\ell/2, \dots, \ell/2 \,. \tag{43}$$

In Fig. 2(a), for $\gamma = 10$, we compare the numerical distribution obtained from an histogram to the theoretical prediction obtaining a good agreement. Since we want to compare this distribution to a discrete one, we normalized the histogram such that the sum over all the heights of the distribution in each bin is equal to one.

On the other hand, the case in which $\gamma \ll 1$ means that measurements are diluted in time and that the information can propagate along the chain. We then have a dynamics dominated by the unitary evolution which may produce an entangled state by propagating the defects generated from the projective measurements. In first approximation, in the limit of $\gamma \ll 1$, we

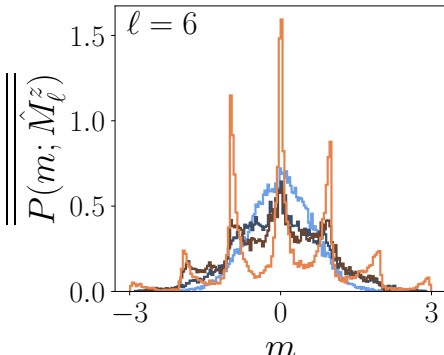 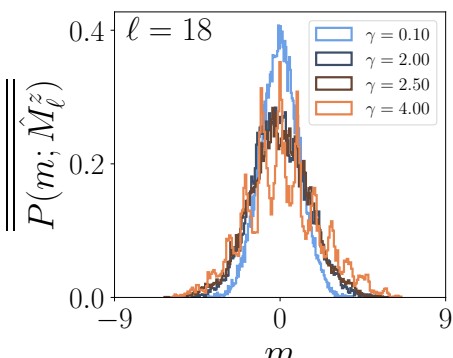

Figure 3: Distribution of the magnetization $\hat{M}_\ell^z$ on a sub-lattice of size $\ell$ centered in the middle of the spin chain in the stationary state $\gamma t \gg 1$. The values of the distribution are extracted form an histogram.

found from the numerics that the local magnetization $m_z(j, \gamma t)$ is distributed in $[-1/2, 1/2]$ with a variance $\sigma^2 \approx 1/16$. From the central limit theorem, we thus find that the subsystem magnetization is distributed as a Gaussian centered in zero with standard deviation $\sigma \sqrt{\ell}$, and thus its probability distribution is given by

$$\overline{\overline{P(m; \hat{M}_\ell^z)}} = \sqrt{\frac{16}{2\pi\ell}} \exp\left[-\frac{16(m_\ell^z)^2}{2\ell}\right]. \tag{44}$$

In Fig. 2(b), for $\gamma = 0.1$, we compare the numerical distribution obtained from an histogram, now normalized such that the integral over the bins is equal to one, to the Gaussian distribution obtaining a good agreement.

Finally, in Fig. 3, for $\gamma = 0.1, 2, 2.5, 4$ and $\ell = 6, 18$ we plot the distributions of the subsystem magnetizations. Qualitatively, as it is suggested by the plot, there is a crossover from a Gaussian distribution to the binomial one. Furthermore, for values of the measurement rate around the critical value of the measurement induced phase transition $\gamma_c \simeq 4$, the distribution starts to develop peaks in correspondence of $m_\ell^z \in -\ell/2, \ldots, \ell/2$ which are the values of the eigenvalues of $\hat{M}_\ell^z$.

In order to study the latter behavior more deeply, in Fig. 4 we plot, for different subsystem sizes, the value of the second moment of the subsystem magnetization, rescaled with the subsystem size, against the measurement rate $\gamma$

$$\mu_2(\hat{M}_\ell^z) = \frac{1}{t_f - t_0} \int_{t_0}^{t_f} \overline{\text{Tr}\{\hat{M}_\ell^z \hat{\rho}_{t,\xi}\}^2} \, dt , \tag{45}$$

where once again $\gamma t_0 = 5$ and $\gamma t_f = 20$. When $\gamma$ is less than 4, so that the dynamics is in the long-range correlated region of the phase diagram, there is a perfect match of the data points. Increasing the value of $\gamma$ we witness spreading of the averages, meaning that we are in a different regime. As a matter of fact, fluctuations of the transverse magnetisation over the stochastic trajectories are extremely favorable indicator to detect a dynamical phase-transition. In particular, the critical value of the measurement rate is located at $\gamma_c \simeq 4$, in perfect agreement to what has been observed by studying the entanglement entropy in Ref. [30].

## 5.2 Ferromagnetic magnetization

We are now going go study the behavior of the ferromagnetic magnetization along $x$ in the stationary state. We can not proceed as in the previous section. Indeed, due to the $\mathbb{Z}_2$ symmetry

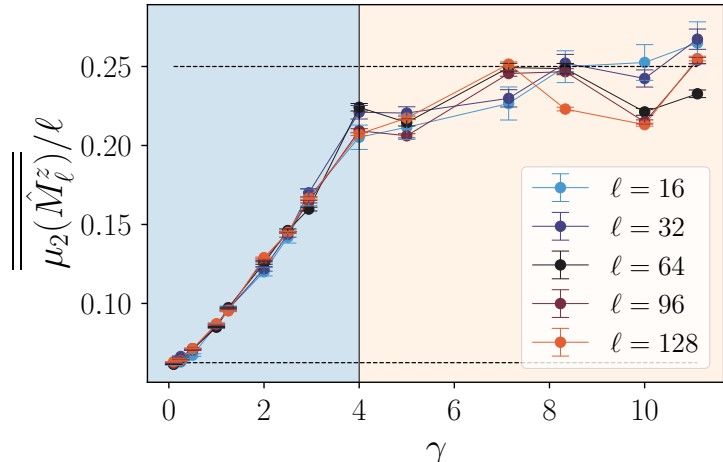

Figure 4: Second moment of the subsystem paramagnetic magnetization $\hat{M}_\ell^z$ rescaled with the size of the subsystem. If $\gamma < 4$ there is a perfect matching between data, after the phase transition we witness a different behavior of the magnetization. Error bars are given by the error of the mean.

of the protocol and of the initial state $\langle \hat{\sigma}_j^x \rangle = 0$ in any site and all the times. This result in a trivial distribution of the ferromagnetic magnetization

$$P_t(m; \hat{M}_\ell^x) = \delta(m) \quad \forall\, t\,. \tag{46}$$

On the other hand, we can consider the quantum probability

$$\mathscr{P}_{t,\xi}(m; \hat{M}_\ell^x) = \text{Tr}\left\{\delta\left(\hat{M}_\ell^x - m\right)\hat{\rho}_{t,\xi}\right\}, \tag{47}$$

and compute the generating function of the cumulants. In particular we studied the fourth cumulant. Despite it has a non trivial a priori evolution, it does not contain any relevant information on the measurement-induced phase transition, due to the fact that we could not have access to sufficiently large subsystems. We present the detailed analysis in the Appendix B.

### 5.2.1 Probability of $M_\ell^{xx}$

In order to overcome the limitations described in the previous paragraph, we considered the full counting statistics over the trajectories of the following observable

$$\hat{M}_\ell^{xx} = \frac{1}{4}\sum_{i \neq j \in \ell} \hat{\sigma}_i^x \hat{\sigma}_j^x\,. \tag{48}$$

To extract information on the spectrum of $\hat{M}_\ell^{xx}$, we rewrite its expression as follow

$$\hat{M}_\ell^{xx} = \frac{1}{4}\sum_{i,j \in \ell} \hat{\sigma}_i^x \hat{\sigma}_j^x - \frac{\ell}{4} = \frac{1}{4}\left(\sum_{i \in \ell} \hat{\sigma}_i^x\right)^2 - \frac{\ell}{4} = \frac{\left(2\hat{M}_\ell^x\right)^2 - \ell}{4}\,. \tag{49}$$

The maximum eigenvalue $\hat{M}_\ell^{xx}$ corresponds to $\ell(\ell-1)/8$ while the minimum is $-\ell/8$. Since we consider an evolution starting from the GHZ state we start from the maximum of value of $\langle \hat{M}_\ell^{xx} \rangle$ and evolve towards a stationary state. In Fig. 5(a) we show the stationary classical

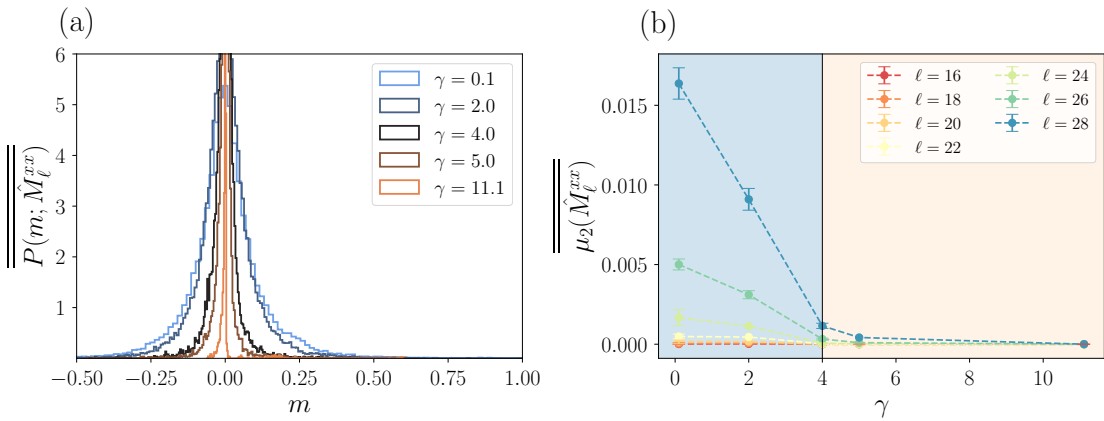

Figure 5: (a) Stationary probability distribution of $\hat{M}_\ell^{xx}$ for $\ell = 28$ integrated for $5 < \gamma t_0 < 20$. (b) Variance of $m_\ell^{xx}$ extracted from the probabilities distributions, the errorbar is given by its fluctuation.

probability $\overline{\overline{P(m; \hat{M}_\ell^{xx})}}$ for a subsystem of size $\ell = 28$. In the case in which $\gamma \gg 1$ the system is not far form an eigenstate of $\hat{M}_\ell^z$ thus the distribution is well described by a $\delta(m)$, we expect thus that all the moments of the distribution in this limit to be equal to zero. On the other hand, decreasing the value of $\gamma$ the distribution transition towards a distribution centered in $m_\ell^{xx} = 0$ with a width that increases decreasing the value of $\gamma$. Indeed, in Fig. 5(b) we plot the width of the aforementioned distribution for different values of the subsystem size, which decreases with the measurement rate $\gamma$. We see that $(\hat{M}_\ell^x)^2$ could witness the measurement induced phase transition since crossing the critical value $\gamma_c \simeq 4$ the width changes dramatically behavior with the subsystem size: in the Zeno-like regime (namely for $\gamma > 4$), the fluctuations are basically suppressed; instead, for $\gamma < 4$ they show a remarkable dependence with the $\ell$, already for relatively small sizes.

As a matter of fact, although this behavior seems not as clean as what we have found for the paramagnetic magnetization, the ferromagnetic fluctuations have the paramount advantage to keep the extensive (with the subsystem size) character only when entering the strongly correlated phase. In other words, while $\mu_2(\hat{M}_\ell^z)$ is expected to show a non-analytic behavior at $\gamma \simeq 4$ in the thermodynamics limit; $\mu_2(\hat{M}_\ell^{xx})$ is not just non-analytic at the transition point, but in addition it clearly characterizes the entire correlated phase already looking at small subsystems.

## 6 Conclusion

The interplay of local measurements and unitary evolution can give rise to phase transitions, manifesting in, e.g., either delocalized, strongly entangled or localized, weakly entangled conditional states.

In this work, we investigated the quantum quench dynamics in a quantum Ising chain under local projective measurements of the paramagnetic magnetization $S_z$.

Very much like in a classical equilibrium situation, when non-commuting observables compete in driving a system accross a quantum phase transition; here the unitary driving and the projective measurements compete in creating or destroying the local order.

In a genuinely statistical sense, different quantum trajectories naturally fluctuate under our dynamical map; this gives rise to non-equilibrium probability distributions of local quantities

which contain signature of paramount yet elusive transitions, going much beyond the simple dynamics of the mean state.

In particular, during the time evolution, by computing the statistics of the expectation values of the system magnetisation in the $z$ direction, we are able to distinguish different regimes, namely different phases. Starting from the strong measurement phase, increasing the imperfection rate, the distribution changes from a bimodal distribution into a Gaussian distribution, the transition point being located at measurements rate $\gamma_c \simeq 4$, in agreement with what have been observed for the entanglement entropy transition [30]. Therefore, our article paves the way for considering second-order cumulants, or even the complete statistics, of quantum averages over sets of quantum trajectories as witnesses of measurement-induced quantum phase transitions.

As a matter of fact, our approach, based on the observation of the statistics of local quantities, is naturally related to what is done in the nowadays experiments. However, especially for devising projective-measurement protocols in the real quantum world, the ultimate challenge, which need to be addressed yet, remains the *post-selection problem*: namely the possibility to experimentally reproduce the same trajectory $\hat{\rho}_{t,\xi}$ many many times, without being affected by the exponentially inefficient measurement-induced post-selection.

# Acknowledgements

The authors acknowledge precious discussions with A. De Luca and X. Turkeshi.

**Funding information** The work has been supported by the ERC under grant agreement n.101053159 (RAVE) and by a Google Quantum Research Award.

# A  Lindbladian dynamics of the averaged state

The projective measurement protocol outlined in the main text relies on the fact that, at every single measurement step, we know which lattice sites are measured, together with the outcomes of the measurements as well.

However, if we do not know whether the lattice site $k$-th is measured and no information about the measurement is retained, then a generic state $\rho$ transforms accordingly to the quantum mechanic prescription as follow

$$\hat{\rho} \to \mathcal{M}_k(\hat{\rho}) = \left(1 - \frac{\gamma\,dt}{2}\right)\hat{\rho} + \frac{\gamma\,dt}{2}\hat{\sigma}_k^z\,\hat{\rho}\,\hat{\sigma}_k^z, \tag{A.1}$$

where $\gamma\,dt$ is the probability that a single site is measured, after a discretization of the continuum time evolution has been applied. Therefore, after a time step $dt$ the entire system with $L$ lattice sites transform according to

$$\hat{\rho} \to e^{-i\,dt\hat{H}}[\mathcal{M}_L \circ \cdots \circ \mathcal{M}_2 \circ \mathcal{M}_1(\hat{\rho})]e^{i\,dt\hat{H}}. \tag{A.2}$$

The discrete protocol in the previous equation can be easily implemented in a Tensor Network algorithm, where each measurement operation $\mathcal{M}_k$ is easily implemented as a transformation of the local tensor in the MPO representation of the mixed state $\hat{\rho}$.

From an analytical point of view, if we are interested in the continuum limit of Eq. (A.2), where $dt \to 0$ with fixed $\gamma$, we can keep the first order terms in the composition of the mea-

surement string, obtaining

$$\mathcal{M}_L \circ \cdots \circ \mathcal{M}_2 \circ \mathcal{M}_1(\hat{\rho}) = \left(1 - L\frac{\gamma\, dt}{2}\right) + \frac{\gamma\, dt}{2}\sum_k \hat{\sigma}_k^z \hat{\rho}\,\hat{\sigma}_k^z + O(dt^2). \qquad (A.3)$$

Combining the previous expansion with the unitary part in the evolution, we finally get the following Lindblad master equation

$$\partial_t \hat{\rho} = -i[\hat{H}, \hat{\rho}] + \frac{\gamma}{2}\sum_{k=1}^{L}\left(\hat{\sigma}_k^z \hat{\rho}\,\hat{\sigma}_k^z - \frac{1}{2}\{\hat{\sigma}_k^z \hat{\sigma}_k^z, \hat{\rho}\}\right), \qquad (A.4)$$

where we used the fact that $(\hat{\sigma}_k^z)^\dagger = \hat{\sigma}_k^z$ and $(\hat{\sigma}_k^z)^2 = 1$.

Eq.s (A.2) and (A.4) describe respectivelly the discrete and the continuous version of the dynamics experienced by averaged state $\overline{\hat{\rho}}$ as well.

In our protocol, the initial state $|\psi_0\rangle\langle\psi_0|$ admit a MPO representation whose local tensors for each lattice site $k$ are

$$\Gamma_k = \begin{bmatrix} |\uparrow\rangle\langle\uparrow| & 0 & 0 & 0 \\ 0 & |\uparrow\rangle\langle\downarrow| & 0 & 0 \\ 0 & 0 & |\downarrow\rangle\langle\uparrow| & 0 \\ 0 & 0 & 0 & |\downarrow\rangle\langle\downarrow| \end{bmatrix} = \frac{1}{2}\begin{bmatrix} 1+\hat{\sigma}^x & 0 & 0 & 0 \\ 0 & \hat{\sigma}^z - i\hat{\sigma}^y & 0 & 0 \\ 0 & 0 & \hat{\sigma}^z + i\hat{\sigma}^y & 0 \\ 0 & 0 & 0 & 1-\hat{\sigma}^x \end{bmatrix},$$

$$(A.5)$$

and both left and right boundary vectors are given by $\vec{l} = \vec{r} = (1,1,1,1)/\sqrt{2}$. Once again, here $|\uparrow\rangle$ and $|\downarrow\rangle$ represents the eigenstates of $\hat{\sigma}^x$ with eigenvalues respectively $+1$ and $-1$. In particular, even under the action of the local transformation $\mathcal{M}_k$ (which does not change the MPO auxiliary dimension), the averaged state remains always an eigenstate of the classical Ising Hamiltonian $H_{xx}$. In other words, the unitary part in Eq. (A.2) does not play any role, and the only contribution to the averaged state evolution comes from the nested application of $\mathcal{M}_k$ on each lattice site. In addition, each single operator in the diagonal MPO $\Gamma_k$, transforms independently.

The local dynamics induced by the nested transformations of $\mathcal{M}_k$ can be easily solved in the Pauli matrix representation of each local state. Indeed, discarding the index $k$ for a sake of clarity, and expanding a generic local density matrix as $\rho = \sum_\mu c_\mu \hat{\sigma}^\mu$, we easily get

$$c_\mu(t) = \sum_\nu \mathbb{M}(t)_{\mu\nu} c_\nu(0), \quad \text{with} \quad \mathbb{M}(t) = \begin{pmatrix} 1 & 0 & 0 & 0 \\ 0 & e^{-\gamma t} & 0 & 0 \\ 0 & 0 & e^{-\gamma t} & 0 \\ 0 & 0 & 0 & 1 \end{pmatrix}. \qquad (A.6)$$

Using this last result with the initial condition in Eq. (A.5) we obtain

$$\Gamma_k(t) = \frac{1}{2}\begin{bmatrix} 1 & 0 & 0 & 0 \\ 0 & \hat{\sigma}^z & 0 & 0 \\ 0 & 0 & \hat{\sigma}^z & 0 \\ 0 & 0 & 0 & 1 \end{bmatrix} + \frac{e^{-\gamma t}}{2}\begin{bmatrix} \hat{\sigma}^x & 0 & 0 & 0 \\ 0 & -i\hat{\sigma}^y & 0 & 0 \\ 0 & 0 & i\hat{\sigma}^y & 0 \\ 0 & 0 & 0 & -\hat{\sigma}^x \end{bmatrix}. \qquad (A.7)$$

The time evolved averaged state is therefore described by $\overline{\rho}(t) = \vec{l}\cdot\prod_{k=1}^{L}\Gamma_k(t)\cdot\vec{r}$, and it relaxes toward the infinite temperature state within the $\mathbb{Z}_2$ symmetry sector with $P = 1$, namely $\overline{\rho}(\infty) = (1+\hat{P})/2^L$.

In addition, the averaged generating function of the moments of $M_\ell^x$ can be easily computed as follow

$$\text{Tr}\{e^{\lambda\hat{M}_\ell^x}\overline{\rho}(t)\} = t\frac{1}{2}\left\{[\cosh(\lambda/2) + e^{-\gamma t}\sinh(\lambda/2)]^\ell + [\cosh(\lambda/2) - e^{-\gamma t}\sinh(\lambda/2)]^\ell\right\}, \quad (A.8)$$

which is expected to be different form the average of the cumulant generating function that has been evaluated in the main text.

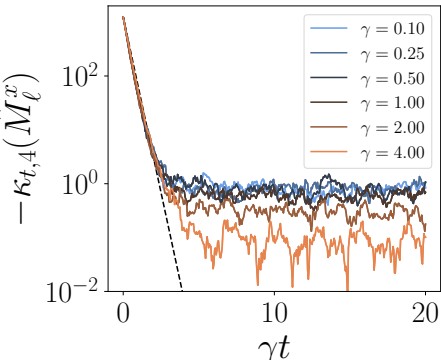
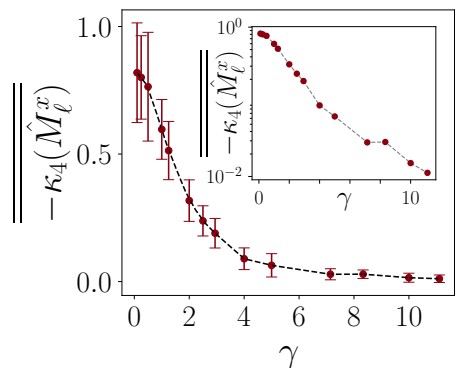

Figure 6: *Non a priori* contribution to the fourth cumulant. Left panel: evolution of the fourth cumulant towards the stationary state. Right panel: time averaged fourth cumulant in the stationary state (after $\gamma t_0 = 5$), the error bars are estimated as the standard deviation of the time average, inset log-linear plot of the mean values. Subsystem size $\ell = 10$.

## B  The full counting statistics

Since we are interested in the statistics of the order parameter $\hat{M}_\ell^x = \frac{1}{2}\sum_{j=1}^\ell \hat{\sigma}_j^x$, in a subsystem of $\ell$ contiguous lattice sites, we do identify $F_\ell(\lambda)$ via

$$\mathrm{e}^{F_\ell(\lambda)} \equiv \langle \mathrm{e}^{\lambda \hat{M}_\ell^x}\rangle, \quad \text{with} \quad K_\ell^n = \partial_\lambda^n F_\ell(\lambda)\big|_{\lambda=0}, \tag{B.1}$$

as the generating function of all cumulants $K_\ell^n$ of the subsystem magnetization. From the large deviation theory we may expect $F_\ell(\lambda) \sim \ell \tilde{F}(\lambda)$ for $\ell \gg 1$, where $\tilde{F}(\lambda)$ is the large deviation function. However, this relies on the extensive behaviour of the cumulants, which is violated in the initial GHZ state. For such reason, it is worth to investigate at the average over the quantum trajectory dynamics of the ratio $\overline{F_\ell(\lambda)}/\ell = \overline{\log\langle \mathrm{e}^{i\lambda \hat{M}_\ell^x}\rangle}/\ell$.

The computation of the subsystem generating function is a very hard task mainly because $\hat{\sigma}^x$ is a nonlocal operator in terms of Majorana fermions. By exploiting the $\mathbb{Z}_2$ symmetry of the measurement protocol, we have

$$F_\ell(\lambda) \equiv G_\ell(\lambda) + \ell \log\cosh(\lambda/2), \tag{B.2}$$

with

$$G_\ell(\lambda) = \log \sum_{n=0}^{\lfloor \ell/2 \rfloor} \tanh(\lambda/2)^{2n} \sum_{j_1 < j_2 < \cdots < j_{2n}}^{\ell} \langle \hat{\sigma}_{j_1}^x \hat{\sigma}_{j_2}^x \cdots \hat{\sigma}_{j_{2n}}^x \rangle, \tag{B.3}$$

where the ordered indexes $\{j_1, \ldots, j_{2n}\}$ are in the interval $[1, \ell]$. Here we decided to highlight the nontrivial part $G_\ell(\lambda)$ of the generating function, whilst the second term in Eq. (B.2) simply gives the infinite temperature contribution. Indeed, we may define the *non a priori* contribution $\kappa_n$ of the cumulants as

$$\partial_\lambda^n F_\ell(\lambda)\big|_{\lambda=0} = \kappa_n + \ell\,\partial_\lambda^n \log\cosh(\lambda/2)\big|_{\lambda=0}, \tag{B.4}$$

so that $\kappa_n \equiv \partial_\lambda^n G_\ell(\lambda)\big|_{\lambda=0}$. The evaluation of $G_\ell(\lambda)$ reduces to the computation of the generic string $\langle \hat{\sigma}_{j_1}^x \hat{\sigma}_{j_2}^x \cdots \hat{\sigma}_{j_{2n}}^x \rangle$. Following Ref. [74], it can be evaluated as the Pfaffian of a skew-symmetric real matrix which explicitly depends on the particular choice of the indices:

$$\langle \hat{\sigma}_{j_1}^x \hat{\sigma}_{j_2}^x \cdots \hat{\sigma}_{j_{2n}}^x \rangle = (-1)^{\mathcal{L}_{j_n}(\mathcal{L}_{j_n}-1)/2}\,\mathrm{pf}\begin{bmatrix} \mathbb{F}_{j_n}^{yy} & \mathbb{G}_{j_n}^{yx} \\ \mathbb{G}_{j_n}^{xy} & \mathbb{F}_{j_n}^{xx} \end{bmatrix}, \tag{B.5}$$

where we used the shorthand notation $\boldsymbol{j}_n \equiv \{j_1, \ldots, j_{2n}\}$ for the full set of indices, and $\mathcal{L}_{\boldsymbol{j}_n} = \sum_{k=1}^{n}(j_{2k} - j_{2k-1})$. The real matrices $\mathbb{F}_{\boldsymbol{j}_n}$ and $\mathbb{G}_{\boldsymbol{j}_n}$ ($\mathbb{F}_{\boldsymbol{j}_n}$ being also skew-symmetric) have dimensions $\mathcal{L}_{\boldsymbol{j}_n} \times \mathcal{L}_{\boldsymbol{j}_n}$ and entries given by [75]

$$(\mathbb{F}_{\boldsymbol{j}_n}^{yy})_{m_p,n_q} = -i\langle \hat{a}_p^y \hat{a}_q^y \rangle + i\delta_{pq} = -i\mathbb{A}_{pq}^{yy} + i\delta_{pq}, \tag{B.6}$$

$$(\mathbb{F}_{\boldsymbol{j}_n}^{xx})_{m_p,n_q} = -i\langle \hat{a}_{p+1}^x \hat{a}_{q+1}^x \rangle + i\delta_{pq} = -i\mathbb{A}_{p+1\,q+1}^{xx} + i\delta_{pq}, \tag{B.7}$$

$$(\mathbb{G}_{\boldsymbol{j}_n}^{yx})_{m_p,n_q} = -i\langle \hat{a}_p^y \hat{a}_{q+1}^x \rangle = -i\mathbb{A}_{p\,q+1}^{yx}, \tag{B.8}$$

$$(\mathbb{G}_{\boldsymbol{j}_n}^{xy})_{m_p,n_q} = -i\langle \hat{a}_{p+1}^x \hat{a}_q^y \rangle = -i\mathbb{A}_{p+1\,q}^{xy}, \tag{B.9}$$

with $\{p,q\} \in [j_1, j_2 - 1] \cup [j_3, j_4 - 1] \cup \cdots \cup [j_{2n-1}, j_{2n} - 1]$ and where the indices $m_p$ and $n_q$ run in $\{0, \ldots, \mathcal{L}_{\boldsymbol{j}_n} - 1\}$, and have the function of shrinking all together the intervals. The knowledge of the Majorana correlation functions together with the representation (B.5) are the basic ingredients to compute the generating function in Eq. (B.3).

We note that, due to the $\mathbb{Z}_2$ symmetry of our system, all the odds cumulants are null. Moreover, for the same reason the second cumulant has a trivial *a priori* evolution since

$$K_{t,2}(\hat{M}_\ell^x) = \overline{\mathrm{Tr}\{(\hat{M}_\ell^x)^2 \hat{\rho}_{t,\xi}\}} - \overline{\mathrm{Tr}\{\hat{M}_\ell^x \hat{\rho}_{t,\xi}\}}^2 = \mathrm{Tr}\{(\hat{M}_\ell^x)^2 \overline{\hat{\rho}_{t,\xi}}\}, \tag{B.10}$$

since the non-linear contribution is equal to zero. The first non-trivial contribution is therefore the fourth cumulant, namely

$$K_{t,4}(\hat{M}_\ell^x) = \mathrm{Tr}\{(\hat{M}_\ell^x)^4 \overline{\hat{\rho}_{t,\xi}}\} - 3\overline{\mathrm{Tr}\{(\hat{M}_\ell^x)^2 \hat{\rho}_{t,\xi}\}}^2, \tag{B.11}$$

where the second term does give a non-linear contribution. In Fig. 6 we plot the time evolution of the *non a priori* part of the fourth cumulant, i.e. $\kappa_{t,4}$, and its time average in the stationary state

$$\overline{\overline{\kappa_4(\hat{M}_\ell^x)}} = \frac{1}{t_f - t_0} \int_{t_0}^{t_f} \overline{\kappa_{t,4}(\hat{M}_\ell^x)}\, \mathrm{d}t, \tag{B.12}$$

with $\gamma t_0 = 5$ and for a subsystem of size $\ell = 10$, and where again $\overline{\overline{(\ldots)}}$ denotes a time average in the stationary configuration. Increasing the value of the measurement rate $\gamma$, we find an exponential decay of the stationary value of the 4-th cumulant towards zero, namely the infinite temperature value. On the other hand, the non-trivial time-evolution does not contain any relevant information on the measurement-induced phase transition. This is probably due to the fact that we could not have access to sufficiently large subsystems. As a matter of fact, the numerical evaluation of the full counting statistics is a very involved procedure, which scales exponentially with the subsystem dimension, thus not allowing to reach thermodynamic relevant sizes.

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
