# Peer review of "Full counting statistics as probe of measurement-induced transitions in the quantum Ising chain"

_SciPost Physics, doi:SciPost Phys. 15, 096 (2023)_

## Round 2 · Referee Report · Anonymous (Referee 1) · 2023-5-1

Strengths

  1. Very active research topic
  2. The chosen system allows a powerful combination of analytic and numerical approaches
  3. Derivations and computations are clearly presented and easy to understand

Weaknesses

  1. The physical meaning of the computed quantity is not clarified
  2. Conclusions are sketchy, implications relevant to ongoing research are not stated
  3. Simplicity of model excludes examination of effects of quantum interactions

Report

This paper considers measurement-induced phase transitions. Instead of focusing on the dynamics of entanglement, the authors consider the effect of continuous measurement on full counting statistics. They chose the Ising quantum spin chain with a zero transverse magnetic field, which is a simple enough system to allow the application of a powerful combination of analytic and numerical tools.
The main weakness of the paper is that the physical (whether theoretical or experimental) significance of the quantity they consider is unclear. Instead of a straightforward construction of full counting statistics, which, as the authors observe, would lead to a rather trivial result due to its linear dependence on the density matrix, the authors chose to average the cumulants of magnetisation over the quantum trajectories which are nonlinear in the density matrix, and then use these results to reconstruct a distribution.
It is entirely unclear what the significance of this distribution is, and how it is related to eventual physics, even at the theoretical level. The paper completely lacks a discussion of this important point.
The second main issue, which seems to be related to the first one is that the conclusions are rather sketchy and descriptive, no deeper physical consequences are drawn from the results. It is not at all clear how these results may be relevant for subsequent work in the field.
In my opinion, clearly, none of the expectations for acceptance in Scipost Physics (groundbreaking discovery; breakthrough on a stumbling block; opening a new pathway; a novel and synergetic link between different research areas) is met. As a result, I cannot recommend the paper for publication in Scipost Physics in its present form, and suggest transferring the paper to Scipost Physics Core, where it can be published once the authors addressed the requested changes.

Requested changes

  1. The authors should clarify the physical content of the full counting distribution they constructed.
  2. The conclusions must be stated clearly, highlighting their significance for subsequent research.
  3. It would also be useful if the authors could discuss how the inclusion of interactions (such as a transverse magnetic field) would affect the results. Does the integrability of such interaction matter?

  • validity: high
  • significance: ok
  • originality: good
  • clarity: high
  • formatting: excellent
  • grammar: excellent

Author:  Mario Collura  on 2023-06-21  [id 3745]

(in reply to Report 1 on 2023-05-01)

see the attached pdf

Attachment:

Ising_order_melting_under_paramagnetic_measures.pdf

---

## Round 2 · Referee Report · Anonymous (Referee 2) · 2023-5-4

Strengths

  • interesting quantity at study in the contect of MIPT

Weaknesses

  • the paper seems rushed out with no clear conclusions or connections to previous works/other observables

Report

The authors want to propose the cumulants of magnetic fluctuations as a different observable to detect MIPT. The idea is nice, and it could also be that this quantity performs better than others (entanglement entropy, purification time, etc). The problem is that they do not do any comparison. The critical gamma ~ 4 they find is not confirmed by any other piece of literature or any other method, for example entanglement entropy scaling or connected correlator. So as it stands the result is not trustable and could be just a crossover.

Requested changes

I suggest to the authors to consider a model where MITP is established (see for example https://arxiv.org/abs/2303.12216, https://arxiv.org/abs/2302.12820) and benchmark their approach there.
Otherwise, to simply compare with other observables known to detect MIPT. Only later the paper can be considered for publication.

---

## Round 3 · Referee Report · Anonymous (Referee 2) · 2023-6-23

Strengths

Same as before

Weaknesses

Previous weaknesses have been solved

Report

I thank the authors to implement the suggestion, the paper is now ready for publication.

---

## Round 3 · Referee Report · Anonymous (Referee 1) · 2023-7-6

Strengths

As before.

Weaknesses

Mostly resolved; c.f. report below

Report

Compared to the previous version, the authors answered weaknesses 1 and 2 by adding more explanatory text to the paper. Specifically, they added sentences clarifying the nature of the quantity computed and extended the conclusion by discussing the relevance of their results for subsequent research.

Weakness 3 was that the simplicity of the model excludes examining the effects of quantum interactions. The authors did answer this in their reply to the referee's comments. I think that discussing this point in the conclusions would eventually be much better and add to the value of the work. I suggest that the authors consider adding their corresponding argument to the conclusions. However, I do not consider this last point a reason to request another revision.

---

## Round 3 · Author Response

\begin{center}
\textbf{First Referee Report}
\end{center}

This paper considers measurement-induced phase transitions. Instead of focusing on the dynamics of entanglement, the authors consider the effect of continuous measurement on full counting statistics. They chose the Ising quantum spin chain with a zero transverse magnetic field, which is a simple enough system to allow the application of a powerful combination of analytic and numerical tools.

{\color{navyblue} We would like to express our appreciation to the referee for accurately summarizing the essence of our article.}

The main weakness of the paper is that the physical (whether theoretical or experimental) significance of the quantity they consider is unclear. Instead of a straightforward construction of full counting statistics, which, as the authors observe, would lead to a rather trivial result due to its linear dependence on the density matrix, the authors chose to average the cumulants of magnetisation over the quantum trajectories which are nonlinear in the density matrix, and then use these results to reconstruct a distribution. It is entirely unclear what the significance of this distribution is, and how it is related to eventual physics, even at the theoretical level. The paper completely lacks a discussion of this important point.

{\color{navyblue} We would like to point out that, apart from a short paragraph in the appendices in which we deal with the cumulants, we did not study the traditional quantum full-counting statistics. Instead, our focus was on computing full counting statistics for the set of quantum trajectories corresponding to the average value of magnetizations. This involved analyzing a probability distribution over a set of classical quantities, specifically traces of operators computed on the quantum trajectories. We apologize for any confusion caused by the lack of clarity on this point and have included additional comments in the amended version of the manuscript to address it.}

The second main issue, which seems to be related to the first one is that the conclusions are rather sketchy and descriptive, no deeper physical consequences are drawn from the results. It is not at all clear how these results may be relevant for subsequent work in the field.

In my opinion, clearly, none of the expectations for acceptance in Scipost Physics (groundbreaking discovery; breakthrough on a stumbling block; opening a new pathway; a novel and synergetic link between different research areas) is met. As a result, I cannot recommend the paper for publication in Scipost Physics in its present form, and suggest transferring the paper to Scipost Physics Core, where it can be published once the authors addressed the requested changes.

{\color{navyblue} First, we thank the referee for having spotted out that in our conclusion, we did not sufficiently stressed the novelty of our approach in analysing a possible transition induced by random measurements. We actually added few sentences to stress the importance of our findings. With this respect, we are firmly convinced that our work deserve the right visibility, and thus to be published in Scipost Physics.}

\begin{center}
\textit{Requested changes}
\end{center}
\begin{enumerate}
\item The authors should clarify the physical content of the full counting distribution they constructed.\\
{\color{navyblue} We addressed the request of the referee in the amended version of the manuscript.}
\\ \ \\
\item The conclusions must be stated clearly, highlighting their significance for subsequent research.\\
{\color{navyblue} We added some more comments in the conclusion section.}
\\ \ \\
\item It would also be useful if the authors could discuss how the inclusion of interactions (such as a transverse magnetic field) would affect the results. Does the integrability of such interaction matter?\\

{\color{navyblue}
We do believe that our approach will be unaffected by the presence of a transverse field in the unitary propagator. As a matter of fact,
see e.g. Phys. Rev. B 105, L241114
(2022), the only things that the presence of the magnetic field will do it is to slightly change the location of the transition between correlated to uncorrelated trajectories.
In this sense, our approach, based on the full distribution function of the local order parameter over the quantum trajectories, would detect the transition as well.}

\end{enumerate}

\begin{center}
\textbf{Second Referee Report}
\end{center}

The authors want to propose the cumulants of magnetic fluctuations as a different observable to detect MIPT. The idea is nice, and it could also be that this quantity performs better than others (entanglement entropy, purification time, etc). The problem is that they do not do any comparison. The critical gamma $\sim$ 4 they find is not confirmed by any other piece of literature or any other method, for example entanglement entropy scaling or connected correlator. So as it stands the result is not trustable and could be just a crossover.

{\color{navyblue} We thank the referee for the comments. We are sorry that it was not clear in the manuscript but the phase transition for $\gamma_c \simeq 4$ is an established result in literature, see for example Phys. Rev. B 103, 224210 (2021).
We added few sentences where we explicitly refer to the fact that our findings are in agreement with what already found, for our specific model, by inspecting at the entanglement transition.
Moreover, in our manuscript we consider the cumulants of the full counting statistics only in an appendix. Indeed, in the rest of the paper we study the probability distribution of the \textit{expectation value} of a quantum observable (thus a classical random variable). }

\begin{center}
\textit{Requested changes}
\end{center}
I suggest to the authors to consider a model where MITP is established (see for example https://arxiv.org/abs/2303.12216, https://arxiv.org/abs/2302.12820) and benchmark their approach there.
Otherwise, to simply compare with other observables known to detect MIPT. Only later the paper can be considered for publication.

{\color{navyblue} As pointed out in the previous response, and also in the manuscript, our model has an established Measured Induced Phase Transition (MIPT). We do think that any further comparison with any other model would not give any additional information to what have been already established with our striking findings, and it goes beyond the scope of our presented work.}

---

## Round 3 · List of Changes

The following sentences have been added/modified accordingly to the referees requests:

\begin{itemize}
\item[pag.3 ---]
``In particular, we investigate how the stationary probability distribution of {\color{red} the averaged values over the set of quantum trajectories of the }magnetizations and its momenta (and cumulant) are affected by the monitoring of local degrees of freedom.''

\item[pag.9 ---]
``{\color{red} We notice that this latter probability distribution contains the information on all the moments $\overline{\expval{\hat{A}}{\psi_\xi}^n}$, which describe the statistical properties of the average of $\hat{A}$ over the set of quantum trajectories.}''

\item[pag.10 ---]
``To be more quantitative, we are going to analyze the behavior of $P_{t}\left(m^z_\ell; \hat M^z_\ell\right)${\color{red}, which we remind is the probability distribution of the averaged value of the subsystem paramagnetic magnetization over the set of quantum trajectories,} in the stationary case for which $\gamma t \gg 1 $ by defining the distribution''

\item[pag.12 ---]
``{\color{red}As a matter of fact, fluctuations of the transverse magnetisation over the stochastic trajectories are extremely favorable indicator to detect a dynamical phase-transition. In particular, the critical value of the measurement rate is located at $\gamma_c \simeq 4$, in perfect agreement to what has been observed by studying the entanglement entropy in Ref.[30].}''

\item[pag.14 ---]
``{\color{red} Therefore, our article paves the way for considering second-order cumulants, or even the complete statistics, of quantum averages over sets of quantum trajectories as witnesses of measurement-induced quantum phase transitions.}''

\end{itemize}

---

## Editorial Decision

published